# The MYC-Regulated RNA-Binding Proteins hnRNPC and LARP1 Are Drivers of Multiple Myeloma Cell Growth and Disease Progression and Negatively Predict Patient Survival

**DOI:** 10.3390/cancers15235508

**Published:** 2023-11-21

**Authors:** Marcel Seibert, Sebastian E. Koschade, Verena Stolp, Björn Häupl, Frank Wempe, Hubert Serve, Nina Kurrle, Frank Schnütgen, Ivana von Metzler

**Affiliations:** 1Department of Medicine, Hematology/Oncology, University Hospital, Goethe University Frankfurt, 60590 Frankfurt, Germanyserve@em.uni-frankfurt.de (H.S.); kurrle@med.uni-frankfurt.de (N.K.); 2German Cancer Consortium (DKTK), Partner Site Frankfurt/Mainz, and German Cancer Research Center (DKFZ), 69120 Heidelberg, Germany; 3Frankfurt Cancer Institute, Goethe-University Frankfurt, 60596 Frankfurt, Germany; 4University Cancer Center Frankfurt (UCT), University Hospital Frankfurt, Goethe University, 60590 Frankfurt, Germany

**Keywords:** multiple myeloma, MYC target pathways, RNA-binding proteins (RBPs), heterogeneous nuclear ribonucleoprotein C (hnRNPC), La Ribonucleoprotein 1 (LARP1), translation

## Abstract

**Simple Summary:**

This study aimed to understand how the oncogenic transcription factor MYC affects cellular processes contributing to cancer. Therefore, we used cell lines derived from multiple myeloma (MM), a malignant plasma cell disorder that is highly dependent on MYC expression. Through quantitative mass spectrometry analysis, we examined how MYC depletion affects the proteome of MM cells. We observed that upon MYC depletion, the levels of the two RNA-binding proteins hnRNPC and LARP1 decreased, suggesting a direct regulation by MYC. Notably, a reanalysis of publicly available data demonstrated that high expression of hnRNPC and LARP1 was linked to poor survival and disease progression in MM patients, suggesting their potential as prognostic markers and therapeutic MYC target proteins. Our findings demonstrate the efficacy of our approach in identifying MYC-regulated target proteins that could potentially serve as predictors of both patient survival and disease progression in MM.

**Abstract:**

Multiple myeloma (MM) is a malignant plasma cell disorder in which the MYC oncogene is frequently dysregulated. Due to its central role, MYC has been proposed as a drug target; however, the development of a clinically applicable molecule modulating MYC activity remains an unmet challenge. Consequently, an alternative is the development of therapeutic options targeting proteins located downstream of MYC. Therefore, we aimed to identify undescribed MYC-target proteins in MM cells using Stable Isotope Labeling with Amino Acids in Cell Culture (SILAC) and mass spectrometry. We revealed a cluster of proteins associated with the regulation of translation initiation. Herein, the RNA-binding proteins Heterogeneous Nuclear Ribonucleoprotein C (hnRNPC) and La Ribonucleoprotein 1 (LARP1) were predominantly downregulated upon MYC depletion. CRISPR-mediated knockout of either hnRNPC or LARP1 in conjunction with redundant LARP family proteins resulted in a proliferative disadvantage for MM cells. Moreover, high expression levels of these proteins correlate with high MYC expression and with poor survival and disease progression in MM patients. In conclusion, our study provides valuable insights into MYC’s role in translation initiation by identifying hnRNPC and LARP1 as proliferation drivers of MM cells and as both predictive factors for survival and disease progression in MM patients.

## 1. Introduction

Multiple myeloma (MM) is a hematological malignancy characterized by the expansion of neoplastic plasma cells within the bone marrow, impaired hematopoiesis, osteolytic bone destruction, and renal failure [1]. Over the last two decades, the introduction and widespread use of several novel therapies have substantially improved patient outcomes [2]. However, even in the era of T-cell redirecting therapies, a cure remains elusive, and the outcome of triple-class refractory patients is limited to an overall survival of twelve months [3].

Overexpression and enhanced activation of c-MYC (referred to as MYC) have been identified as one of the major key drivers of MM and are associated with an aggressive phenotype of the disease [4], resulting in enhanced cell growth, proliferation, energy production, and anabolic metabolism [5,6]. Over the last 20 years, several approaches to combating MYC have been investigated, mainly intervening at the level of MYC expression or MYC activation about binding to the respective E boxes or MAX [7,8,9]. Notably, a dominant negative of MYC, Omomyc, has been developed and primarily utilized for genetically driven research on MYC. Recently, a mini-protein compound with cell-penetrating properties, OMO-103, has entered a phase-1 clinical trial for patients with solid malignancies [10]. Despite this promising approach for direct MYC targeting, the development of a clinically applicable molecule modulating MYC activity is still challenging [10,11]. Thus, other approaches that bypass this roadblock are needed to develop attractive therapeutic options for MM patients. One approach is to identify the specific downstream transcription profile induced by MYC to target key effector pathways. Here, genes encoding for ribosomal proteins, ribosomal RNA, and proteins crucial for the set-up of translation initiation were already identified [12]. Inhibitors of translation initiation, such as rocaglate scaffold inhibitors, were shown to inhibit the oncogenic MYC-driven transcriptional program and abrogate myeloma growth in preclinical in vivo MM models [13]. However, while blocking this program is associated with the depletion of specific critical proteins, which is undesirable for MM development and survival, the exact mechanism of MYC-driven translation regulation remains unknown. In this work, we aimed to elucidate the specific protein network regulated by MYC in MM using a SILAC-based MS approach. Moreover, we examined the impact of the thereby identified putative target proteins on myeloma growth and patients’ outcomes.

## 2. Materials and Methods

### 2.1. Cell Culture

Human multiple myeloma (MM)-derived cell lines (RPMI8226, LP1, and OPM2) were obtained from the DSMZ (Deutsche Sammlung von Mikroorganismen und Zellkulturen GmbH, Braunschweig, Germany) and were cultured in RPMI-1640 medium (Gibco/Thermo Fisher Scientific, Darmstadt, Germany #21875034), supplemented with 10% (*v*/*v*) FCS (Merck, Darmstadt, Germany), 100 U/mL penicillin, and 100 μg/mL streptomycin (1× P/S) (Thermo Fisher Scientific, Darmstadt, Germany #15140122). Cell lines were chosen due to different patterns of MYC abnormalities (Table 1).

Human bone marrow stromal cell-derived HS5 cells (obtained from ATCC (American Type Culture Collection, Manassas, VA, USA)) and for lentivirus production, Lenti-X 293T cells (referred to as HEK293T) (Takara Bio Europe SAS, Saint-Germain-en-Laye, France, #Z2180N) were cultured in high glucose DMEM 4.5 g/L D-glucose medium (DMEM 4.5 g/L D-glucose, Gibco/Thermo Fisher Scientific, #41965-039), supplemented with 10% (*v*/*v*) FCS (Sigma-Aldrich, St Louis, MO, USA), and 1× P/S. All cell lines were cultured at 37 °C in a humidified, 5% CO_2_ incubator. All used MM-derived cell lines and the HS5 cell line stably express spCas9 generated beforehand by lentiviral transduction [16] using the plasmid pLentiCas9-Blast (Addgene, Watertown, MA, USA #52962).

### 2.2. Antibodies

For Western blot detection, all antibodies were diluted 1:1000 in TBST (TBS buffer B) (Zytomed Systems, Bargteheide, Germany, #ZUC066), containing 0.05% (*v*/*v*) Tween-20 (Carl Roth GmbH, Karlsruhe, Germany, #9127.1) and 0.5% (*v*/*v*) NaN_3_ (Sigma, #S2002), except for the GAPDH and β-Tubulin antibodies, which were applied in a 1:20,000 dilution: 4EBP1 (Cell signaling Technology (C.S.T.), Frankfurt, Germany, Rabbit #9452), GAPDH (Abcam (6C5), Mouse #ab8245), HK2 ((C.S.T.) (C64G5) Rabbit mAb #2867), hnRNP C1/C2 ((C.S.T.) (D6S3N), Rabbit mAb #91327), LARP1 ((C.S.T.), Rabbit #14763), MYC ((C.S.T.) (D84C12) Rabbit mAb #5605), and β-Tubulin (ProteinTech, Planegg-Martinsried, Germany, Mouse #66240-1-Ig)

### 2.3. Cell Viability and Apoptosis Measurements

MTT-Assay: RPMI8226 cells were treated with the tetrazolium salt 3-(4,5-dimethylthiazol-2-yl)-2,5-diphenyltetrazolium bromide (MTT). The amount of formazan formed was quantified by the absorbance at 570 nm. A total of 2 × 10^5^ RPMI8226 cells/mL were transferred into a well of a 96-well plate (as five technical replicates) in a total volume of 90 µL. For each measurement, 10 µL MTT solution was added to one well (5 mg MTT salt/mL (12 µM) (Carl Roth GmbH, Karlsruhe, Germany, #4022.3)), diluted in 1× PBS, and incubated for 4 h at 37 °C and 5% CO_2_. The reaction was terminated by adding 100 µL SDS solubilization solution (10% SDS, diluted in distilled water with 0.01 M HCl) and incubation at 37 °C with 5% CO_2_ prior to measurement.

Apoptosis measurements: the induction of apoptosis leads to cell surface phosphatidyl-serine exposure, which was stained by a fluorophore (APC)-coupled AnnexinV (AnX) (BD Biosciences, Heidelberg, Germany, #550475). A total of 1 × 10^5^ cells were washed with cold 1× PBS and resuspended in 100 µL Ca^2+^-containing 1× AnX-binding buffer (BD Biosciences, USA #556454) to prime cells for AnX binding. A quantity of 100 µL cell suspensions was stained after 30 min incubation at 4 °C in the presence of 5 µL APC-AnX. The level of apoptosis correlates with APC fluorescence and was determined by flow cytometry (BD LSRFortessaTM Cell Analyzer (BD Biosciences, Franklin Lakes, NJ, USA)). GFP-positive MYC knockout cells were gated, and the percentages of AnX-positive knockout cells were normalized to the percentage of AnX-positive non-target control (NTC) cells.

### 2.4. Cell Modification, CRISPR/spCas9-Mediated Gene Knockout

Gene knockout MM-derived cells were generated by lentiviral transduction of 0.25 × 10^6^ spCas9-expressing MM cell lines in 1 mL standard RPMI1640 medium with pLentiCRISPRv2_ΔspCas9 encoding sgRNAs targeting MYC, hnRNPC, or LARP proteins (LARP1/4/4B). The expression of sgRNAs was coupled to a fluorescent marker expression (Table 2). Gene knockout of MYC in HS5 cells was achieved by lentiviral transduction of 100,000 spCas9-positive HS5 cells, seeded 24 h in advance. We designed all sgRNAs using the Benchling software package (Benchling [Biology Software] (2023), retrieved from https://benchling.com (accessed on 14 April 2021). All oligonucleotides were obtained from Sigma-Aldrich and cloned in pLentiCRISPRv2_ΔspCas9, which is derived from the commercially available pLentiCRISPRv2 plasmid (Addgene #52961), in which the cDNA encoding spCas9 was removed and the cDNA encoding the puromycin resistance cassette was replaced by a fluorescent marker-encoding cDNA (Table 2). The cloning of sgRNAs was performed according to the Golden Gate protocol [17] into the BsmBI site. Vectors carrying a non-target control (NTC) sequence was used as controls (Table 2).

Cloned plasmids were verified by Sanger sequencing, and HEK293T cells (Takara Bio Europe SAS, #Z2180N) were used to produce lentiviral particles [16] using the packaging plasmids pMD2.G (Addgene, #12259) and psPAX2 (Addgene, #12260). Successful transduction was verified by flow cytometric analysis for E2C, BFP, or GFP fluorescence (BD LSRFortessa^TM^ Cell Analyzer (BD Biosciences, USA)). Twenty-four hours after transduction, media were replaced with fresh standard media, and cells were expanded until the days of the experiments.

### 2.5. Competitive Cell Proliferation Assays

The competitive proliferation assay was used to evaluate the effect on cell proliferation caused by gene depletions. Fluorescent knockout (KO) cells were cultured in the presence of untransduced wild-type (WT) cells. The relative cell proliferation of fluorescent knockout cells was compared to that of the corresponding competitive WT cells. Fluorescent and WT cells were mixed in a 50:50 ratio and cultured in the appropriate standard medium for several days. Cell density was adjusted every second day to 0.35 × 10^6^ cells/mL. The relative proportion of fluorescent cells over time was determined every second day by flow cytometry (BD LSRFortessa^TM^ Cell Analyzer (BD Biosciences, USA)).

### 2.6. Stable Isotope Labeling with Amino Acids in Cell Culture (SILAC)

SILAC was conducted to metabolically label proteins by replacing two essential amino acids (arginine and lysine) with corresponding isotopic-labeled amino acids in a dialyzed cell culture medium (heavy- or medium-labeled). SILAC RPMI Medium w/o arginine and lysine, supplemented with L-glutamine (Silantes, Germany #283001200), supplemented with 10% (*v*/*v*) dFCS (Bio&SELL GmbH, Germany), and 1× P/S (Thermo Fisher Scientific, USA). “Light”-labeled SILAC Medium contained 0.219 mM ^12^C_6_-^14^N_2_-lysine and 1.137 mM ^12^C_6_-^14^N_2_-arginine. “Medium”-labeled SILAC Medium contained 0.219 mM D_4_-lysine and 1.137 mM ^13^C_6_-arginine. “Heavy”-labeled SILAC Medium contained 0.219 mM ^13^C_6_-^15^N_2_-lysine and 1.137 mM ^13^C_6_-^15^N_4_-arginine (Eurisotop, Saint-Aubin, France). “Medium”- and “heavy”-labeled SILAC Medium was supplemented with 2 mM proline (Carl Roth GmbH, Karlsruhe, Germany #1713.1).

Cells were cultured for two weeks at 37 °C and 5% CO_2_. Cells that received “light”-labeled amino acids were used as NTC cells, and MYC knockout cells, which were generated upon transduction with two different MYC-targeting sgRNAs (sgRNA(1) and sgRNA(2)), received “medium”- and “heavy”-labeled amino acids, respectively. Lentiviral particles for transduction were prepared in HEK293T cells, which were cultured in the respective SILAC medium. Two days after transduction, the transduced cells were separated into three technical replicates for cell lysis. Relative quantification of changes in protein expression after MYC knockout in the MM-derived RPMI8226, LP1, and OPM2 cells, as well as in HS5 control cells, was calculated by mixing lysates obtained from SILAC of two isotopic-labeled MYC knockout cell lysates (“heavy”- and “medium”-labeled) with “light”-labeled non-target control (NTC) cell lysates in a ratio of 1:1:1 (of three technical replicates). Cells were lysed in MS lysis buffer (50 mM Tris HCl pH8 (Carl Roth GmbH, #9090.3), 150 mM NaCl (Riedel-de-Haёn, #31434), 0.5% NP-40 (Sigma-Aldrich, #NP40S), 5 mM NaF (Sigma-Aldrich, #S7920), 1:50 EDTA-free protease inhibitor cocktail (cOmplete TM, Roche Diagnostics GmbH, #05056489001)). Three times, LD-sample buffer (Thermo Fisher Scientific, #NP0008) containing 200 mM HPLC-grade DL-Dithiothreitol (DTT, Sigma-Aldrich, #D0632) was added, and samples were boiled for 5 min at 95 °C.

### 2.7. Mass Spectrometry and Data Analyses

To identify MM-specific MYC-regulated proteins, the three technical replicates of 1:1:1 lysate mixtures of each cell line were separated by SDS–PAGE using precast Bis-Tris minigels (NuPAGE Novex 4–12%, Life Technologies/Thermo Fisher Scientific). The separated proteins were stained with Coomassie Brilliant Blue (SERVA Electrophoresis GmbH, Heidelberg, Germany) and cut into 23 slices. The proteins in each slice were reduced with DTT (Sigma-Aldrich) and alkylated with iodoacetamide (Sigma-Aldrich) before being digested with trypsin overnight (SERVA Electrophoresis GmbH). The peptides were then extracted from the gel matrix and analyzed by liquid chromatography/tandem mass spectrometry (LC-MS/MS) using a quadrupole-Orbitrap hybrid mass spectrometer (Q Exactive, Thermo Fisher Scientific) coupled to an EASY n-LC 1000 HPLC system (Thermo Fisher Scientific). The samples were first desalted on a trap column (20 × 0.1 mm; packed in-house with ReproSil-Pur 120 C18-AQ, 5 μm (Dr. Maisch GmbH, Ammerbuch, Germany)) at 5 μL/min in loading buffer [2% (*v*/*v*) ACN, 0.1% FA] and then separated on an analytical column (320 × 0.075 mm; packed in-house with ReproSil-Pur 120 C18-AQ, 1.9 μm (Dr. Maisch GmbH, Ammerbuch, Germany)) using an 80 min linear gradient from 5% to 42% buffer B [95% (*v*/*v*) ACN, 0.1% FA] over buffer A (0.1% FA) at a flow rate of 300 nL/min. A data-dependent acquisition scheme was used to analyze eluting peptides. The 20 most abundant precursor ions with charge states of 2 to 64 were selected in an isolation window of *m*/*z* 2.0 and subjected to higher-energy collisional dissociation (HCD) at an NCE of 30%. The MS survey spectra ranged from *m*/*z* 350 to 1600, with a resolution of 70,000 FWHM at *m*/*z* 200. For the MS/MS product ion spectra, the resolution was set to 17,500 FWHM at *m*/*z* 200. The AGC target values and maximum injection times for MS and MS/MS were 1 × 10^6^ at 50 ms and 1 × 10^5^ at 54 ms, respectively. Fragmented ions were also excluded from isolation for 25 s.

The LC-MS/MS data were processed using the MaxQuant software (version 1.6.0.1, MPI for Biochemistry) [18]. The UniProtKB/Swiss-Prot human database, which contained 88,993 protein entries (downloaded November 2016), supplemented with 245 commonly encountered contaminants, was used for database searching with the Andromeda search engine [18]. Precursor and fragment ion mass tolerances of 6 ppm and 20 ppm, respectively, were applied after initial recalibration. Variable modifications included protein N-terminal acetylation and methionine oxidation, while cysteine carbamidomethylation was considered a fixed modification. Peptides shorter than seven amino acids were discarded, and up to two missed tryptic cleavages were permitted. The false discovery rate (FDR) was set to 1% for both peptide and protein levels using a forward and reverse decoy database approach. SILAC quantitation was performed by setting the multiplicity to two for double labeling (Lys + 0/Arg + 0 and Lys + 8/Arg + 10), and a minimum of two ratio counts were required for peptide quantitation. Both the “match between runs” and “re-quantify” features of MaxQuant were enabled.

The MaxQuant output data were further analyzed using Perseus software (version 1.6.0.7, MPI for Biochemistry) [19]. Hits from the decoy database search and potential contaminants were removed, and the SILAC ratios were log_2_-transformed (log_2_ (fold change) values). To identify regulated proteins in each cell line, SILAC ratios of light- (NTC) to medium-/or heavy-labeled proteins (MYC knockout by sgRNA(1) or sgRNA(2)) were calculated. Threshold values to identify regulated proteins were adjusted according to measured log_2_ (fold change) values of the known MYC target proteins in the literature (see Appendix A). To statistically evaluate changes in protein expression upon knockout of MYC, a one sample t-test of log_2_-transformed SILAC ratios was conducted, and Benjamini–Hochberg FDR < 5% was applied to adjust the *p*-values. The threshold for *p*-values calculated by *t*-tests was set to <0.05 (–log_10_ *p*-value > 1.0). For pathway analysis, regulated proteins were listed according to the adjusted log_2_ values and used for overrepresentation analysis (ORA). ORA was used to determine whether certain pathways were enriched in the set of proteins that were more abundant (enriched) than statistically expected, as described in Boyle et al. (2004) [20], revealing differential regulations in response to MYC depletion. The functional enrichment analysis of each protein group was performed using g:Profiler (version e106_eg53_p16_65fcd97) [21] with the Benjamini–Hochberg FDR multiple testing correction method applying a significance threshold of 0.05. To visualize protein and pathway enrichment, we used the Enrichment Map application (v3.3.6) for Cytoscape software (v3.9.1) [22,23], described in Appendix A.

### 2.8. Cell Lysis, SDS-PAGE, and Western Blot

For protein lysates, 1 × 10^6^ cells were lysed by adding 50 µL of SDS lysis buffer (100 mM Tris HCl pH8 (Carl Roth GmbH, #9090.3), 150 mM NaCl (Riedel-de-Haёn, #31434), 10 mM EDTA pH8 (PanReac AppliChem, Darmstadt, Germany, #A4892,0500), 10% SDS (MP Biomedicals, Irvine, CA, USA, #04811033-CF), freshly supplemented with protease inhibitors (cOmplete™, Mini, EDTA-free Protease Inhibitor Cocktail, Roche/Merck, #11836170001). Lysates were cleared by centrifugation. Protein concentrations were determined by the Lowry assay (DC^TM^ Protein Assay Reagents A and B, Biorad, Hercules, CA, USA) according to the manufacturer’s instructions. Equal protein amounts of the lysates were analyzed by SDS–PAGE and Western blot.

### 2.9. Biomarker Analysis

Publicly available RNA-Seq expression data (normalized TPM values) and clinical data from the MMRF CoMMpass study (NCT01454297) [24] release IA19 were retrieved from the GDC data portal using R 4.2.2 and TCGAbiolinks 2.25.3 [25]. Log_2_-transformed expression data were correlated using the Spearman rank coefficient. For survival analyses, expression data for indicated genes was dichotomized into low (below median) and high (above median). Right-censored overall survival was estimated with the Kaplan–Meier method, and the log-rank test implemented in R’s survival 3.5-3 package was used to test for differences in survival between groups. Publicly available normalized microarray expression and associated metadata were retrieved from GEO (GSE6477) [26], and probe identifiers were mapped to ENSEMBL gene identifiers using hgu133a.db 3.13.0. *p*-values for differences in log_2_-transformed expression values between newly diagnosed MM and healthy plasma cells were obtained using an unpaired Wilcox signed rank test. Conditional posterior parameters for the interaction between the target gene and disease stage (normal plasma cell, monoclonal gammopathy of unknown significance, smoldering multiple myeloma, newly diagnosed multiple myeloma, and relapsed multiple myeloma) fitted to log_2_ expression values were obtained from a Bayesian ordinal regression analysis (covariates: target gene as a nominal factor, disease stage as a monotonic ordered predictor, default non-informative priors, and MCMC sampling parameters) using brms 2.19.0.

## 3. Results

The most common event that triggers neoplastic transformation in MM is the deregulation of oncogenic MYC, bringing MYC into focus as a therapeutic target protein. However, direct targeting of MYC for MM treatment is still a clinical challenge. Due to its important role in MM disease progression, we, therefore, analyzed MM-specific MYC-regulated pathways to search for alternative treatment targets by determining proteomic responses upon MYC knockout. Since MYC expression in MM-derived cells leads to increased expression of various genes involved in cell proliferation, depletion of MYC was expected to result in a dramatic reduction in cell proliferation and viability. Therefore, in a first step to determine an optimal time point for mass spectrometry measurements while avoiding overlapping secondary effects, we validated the CRISPR/spCas9-mediated knockout of MYC upon transduction with two different sgRNA in RPMI8226 cells and characterized them about cell proliferation and apoptosis induction (Figure 1).

After two days, MYC expression was reduced by sgRNA(1) and sgRNA(2) to 2.58% and 1.55%, respectively (Figure 1A,B). Furthermore, MYC-depleted RPMI8226 cells exhibit significantly reduced metabolic activity as early as 24 h after transduction (NADH/NADPH concentration, used as a surrogate parameter for proliferation and viability, Figure 1C), suggesting that cell proliferation is considerably reduced upon MYC depletion. This was consistent with an increasing proportion of AnnexinV-positive apoptotic cells up to day 5 post-transduction (Figure 1D). Yet, the proportion of apoptotic cells upon MYC depletion is consistently below 5% until day 3 (Figure 1D). Based on these results, we subsequently analyzed the cellular responses to MYC depletion using a SILAC proteomics approach two days after transduction, a time point at which 95% of cells are viable despite MYC depletion.

### 3.1. Identification of Potential MYC Target Proteins by SILAC and Mass Spectrometry

The three MM-derived cell lines RPMI8226, LP1, and OPM2, which were chosen due to the expression of both abnormal and normal MYC alleles, thereby capturing the diverse range of MYC expression patterns observed in MM patients (Table 1) [14,15], were transferred to SILAC medium 2 weeks before transduction with either non-target-control (NTC) sgRNA as control or MYC-targeting sgRNA(1) and sgRNA(2). For quantitative proteomics, lysates of NTC and knockout cells were collected after 48 h, followed by MS to characterize proteomic changes in MYC-depleted cells. Protein regulations upon MYC knockout of three MM-derived cell lines (RPMI8226, LP1, and OPM2) were compared to those of HS5 stromal cells (Appendix A). Control lysates were prepared for Western blotting, revealing successful MYC depletion in cells used for MS measurements (Appendix A). As shown in volcano plots (Figure 2A–D and Appendix A), proteomic changes upon depletion of MYC (compared to NTC) were identified in all cell lines, expressed as mean log_2_ fold change (FC) values. All proteins that were measured as downregulated and exceeded or fell below a previously analytically determined log_2_ (FC) value are marked in red. Quantified proteins, which were further analyzed based on the enrichment analysis (ORA), such as Eukaryotic Translation Initiation Factor 4E-Binding Protein 1 (4EBP1), La-Ribonucleoprotein 1 (LARP1), and Heterogeneous Nuclear Ribonucleoprotein C (hnRNPC), as well as the two known and well-established MYC targets Hexokinase 2 (HK2) and Y-Box Transcription Factor (YB-1), are depicted in the respective volcano plot.

### 3.2. Investigation of MM-Specific MYC-Driven Pathways

To investigate MM-specific MYC-driven pathways, we initially conducted a thorough literature review of established MYC target protein data. This step was crucial to ensuring the reliability and reproducibility of our chosen datasets (Appendix A). In summary, our determined log_2_ [FC] values for protein regulations upon MYC knockout are consistent with previously published findings [27], confirming independent data reproducibility for investigations into MYC target pathways. Therefore, the data were first used to determine which proteins are exclusively downregulated in MM-derived cell lines and not in HS5 cells, which were used as non-MM control cells. The threshold for log_2_ [FC] values for significantly downregulated proteins was set to <8 fold (=log_2_ [FC] < (−0.32)) and –log_10_ [*p*-value] > 1, as performed previously during mass spectrometry data validation (Appendix A). All identified proteins that were regulated by both sgRNAs targeting MYC were ordered into commonly regulated proteins of different cell lines using a Venn diagram (Figure 3).

Potential MYC target proteins that were identified as exclusively downregulated upon MYC depletion in MM-derived cell lines (with downregulated proteins in HS5 cells excluded (Figure 3)) were used for overrepresentation analysis (ORA). We analyzed 911 proteins, and enriched proteins were subsequently grouped into target pathways according to the REACTOME database [28], revealing a cluster of proteins associated with the regulation of translation initiation (Appendix A). Indeed, this analysis confirms previously determined target pathways of MYC, which are associated with the control of protein synthesis [12,13,29,30]. To identify specific MYC target proteins for further validation, we then conducted exploratory data analysis (Appendix A), revealing certain sub-processes of translation initiation that are specifically affected in MYC-depleted MM cell lines. This identified several specific candidates (Table 3), including small and large ribosomal subunits (RPS and RPL), which were excluded from further analysis.

Among the regulated candidates were mainly eukaryotic initiation factor (EIF) family members that regulate the initiation of translation. We followed up on Western blot validation of the results by assessing the protein expression of the known MYC target EIF 4E Binding Protein (EIF4EBP1, or 4EBP1) as a positive control in MYC-depleted RPMI8226 and LP1 cells. In addition, we extended this investigation to explore the functionally related La Ribonucleoprotein 1 (LARP1), which was identified through our approach and is known to participate in the translation regulation of specific mRNA translations [31,32] but is not part of the translation initiation pathway defined in REACTOME (Figure 4).

In RPMI8226 cells, Western blot analysis revealed that both translation regulatory proteins involved in specific mRNA translation were downregulated to 40.9% (4EBP1) and 53.5% (LARP1) upon MYC depletion. In LP1 cells, 4EBP1 was downregulated to 25.2% and LARP1 to 36.1% (Figure 4). The results demonstrated that in MM cells, MYC serves as a regulator of protein synthesis, in particular by the identification of the two proteins 4EBP1 and LARP1, both of which are involved in the regulation of mRNA translation [32,33]. This provides valuable insight into the mechanism by which MYC regulates translation.

### 3.3. Deregulation of MYC-Driven Translation in MM

Since we confirmed the expression of the two mRNA-translation-regulating proteins 4EBP1 and LARP1 as regulated by MYC, we asked whether RNA-binding and mRNA-regulation in general are MYC-driven functions. As shown in Figure 5A, in MYC-depleted cells, the ORA using Molecular Function (MF) data from gene ontology (GO) revealed an enrichment of proteins involved in “nucleic acid binding” (GO:0003676). As exemplary results showed for LP1 cells, 79.8% of those nucleic acid-binding proteins, which were downregulated upon transduction with MYC-targeting sgRNA(2), belong to RNA-binding proteins (RBPs), including small and large ribosomal subunits (RPS and RPL), and mRNA processing and translation factors (Figure 5B). As shown in Figure 5C,D and highlighted for the MM-derived cell lines RPMI8226 and LP1 cells in Figure 2 and Appendix A, from these RBPs, the Heterogeneous Ribonucleoprotein Particle C (hnRNPC) was further analyzed due to its low log_2_ [FC] values determined for RPMI8226 and LP1 (Appendix A), revealing considerable strong downregulation upon MYC depletion.

Indeed, the expression of hnRNPC was significantly decreased to 29.7% in MYC-depleted LP1 cells. In MYC-depleted RPMI8226 cells, the hnRNPC expression was decreased to 79.4%, suggesting that hnRNPC is less affected by MYC depletion in this cell line (Figure 5C,D). However, we tested the proliferative effect of a hnRNPC knockout in RPMI8226 and LP1 cells in a competitive proliferation assay using two different sgRNA targeting hnRNPC (sgRNA(e3) and sgRNA(e3i)) (Figure 6A). Depletion of hnRNPC resulted in a significant decrease in the percentage of GFP-fluorescent KO cells, demonstrating diminished cell proliferation in a time-dependent manner in both cell lines. Additionally, we analyzed the competitive proliferation of RPMI8226 and LP1 cells upon knockout of the identified RBP LARP1, as well as of other proteins of the LARP family (LARP4 and LARP4B) (Figure 6B). The individual knockout of LARP proteins using two sgRNAs targeting one LARP gene showed limited effects on cell proliferation, indicating the presence of redundancies in their functions. However, competitive growth analysis of RPMI8226 cells upon simultaneous (triple) LARP protein knockout resulted in a considerable growth disadvantage, revealing that LARP proteins are essential for MM cell survival (Figure 6C).

Altogether, these findings suggest that MYC-regulated processes involving hnRNPC and LARP proteins are essential for the in vitro proliferation of MM cell lines. Due to these observations, we next aimed to determine whether hnRNPC, LARP1, and 4EBP1 (EIF4EBP1) are deregulated in MM patients.

Therefore, we analyzed publicly available data sets from patients with MM at initial diagnosis. First, we found a highly significant correlation between MYC expression and expression of 4EBP1, LARP1, and hnRNPC in primary material from 762 newly diagnosed MM patients in the MMRF CoMMpass study (NCT01454297) [24] (Figure 7A). To look at the prognostic significance in terms of the survival of patients with MM, we performed a Kaplan–Meyer analysis on these patients. The results indicated a clear and significant survival disadvantage for patients with high 4EBP1, LARP1, or hnRNPC expression (Figure 7B). Additionally, we found expression of MYC, 4EBP1, LARP1, and hnRNPC to be significantly increased in primary samples from newly diagnosed multiple myeloma (NDMM) patients compared to plasma cells (PC) from healthy donors (Figure 7C). Regression analysis showed that expression of MYC, 4EBP1, LARP1, and hnRNPC increased continuously in primary samples with an advancing disease stage (Figure 7D). Collectively, these findings suggest that MYC, 4EBP1, LARP1, and hnRNPC share a common, deregulated expression profile that correlates with worsening disease stage and negative prognostic impact in MM.

## 4. Discussion

Multiple myeloma (MM) is a complex, frequently MYC-driven, hematological malignancy with a poorly understood etiology and inevitable progression. In this study, we aimed to gain insights into cellular pathways affected by MYC, a critical transcription factor that plays a key role in the proliferation and metabolism of MM cells. Despite being an excellent candidate for therapeutic targeting, however, it remains a challenge to develop drugs targeting MYC directly [34]. To identify yet undescribed MYC-regulated proteins and pathways that could serve as indirect therapeutic options, we utilized mass spectrometry (MS)-based proteomics upon MYC depletion (Figure 2, Appendix A). This will ultimately offer potential therapeutic strategies for the treatment of MM. To avoid MYC target proteins, which are generally downregulated upon MYC depletion and are less MM-specific, in our analysis we excluded all potential MYC target proteins determined in non-MM stromal HS5 control cells (Figure 3).

Our overrepresentation analysis (ORA) of downregulated proteins primarily revealed MYC targets associated with protein synthesis/translation (Appendix A), thereby confirming a strong correlation between MYC expression and ribosomal biogenesis, as previously observed in over 1000 cell lines from the Cancer Cell Line Encyclopedia (CCLE) database [13]. While most MYC-driven cancer cells are known to share basic alterations in translation and ribosome biogenesis [12,13,29,30], particularly in MM-derived cell lines, a strong correlation between *MYC* expression and translation processes has been observed [13]. This correlation is also evident in tumor cell-derived gene expression data from MM patients [13], underscoring MYC’s specific influence on gene expression in MM. Conversely, depletion of MYC in colorectal tumors results in significant impairment of global translation [35]. However, due to the limited understanding of the exact mechanism behind MYC-driven translation regulation, by using REACTOME data, our in-depth analysis of regulated proteins in MYC-depleted MM-derived cell lines uncovered specific proteins intricately regulated by MYC (Table 3). Interestingly, in addition to more general factors being involved in translation, such as structural constituents for ribosome biogenesis, our combined mass spectrometry data analysis of MM-derived cell lines using REACTOME and Gene Ontology (GO) data together identified hnRNPC and LARP1 as MYC-regulated RNA-binding proteins. Notably, many proteins of the LARP protein family seem to be similarly involved in MYC-driven translation regulation, as has already been demonstrated for the translation regulator 4EBP1 [36].

As shown in Figure 4, Western blot analysis verified reduced expression of 4EBP1 in MYC-depleted RPMI8226 and LP1 cells, suggesting MYC’s role in regulating modulators of translation initiation. This MYC-driven 4EBP1 expression corroborates earlier findings that showed that MYC binds to control regions of the *4EBP1* gene [36]. Notably, this known role of MYC in regulating translation initiation gains further depth with the discovery of MYC’s influence on LARP proteins, particularly LARP1 in the MM-derived cell lines RPMI8226 and LP1 (Figure 4B). Interestingly, previous reports underscore the active regulation of translation initiation by a link between MYC and the Mammalian Target Of Rapamycin Complex 1 (mTORC1) [37,38,39,40]. Thus, it has been shown that this kinase complex phosphorylates and regulates both 4EBP1 [41] and LARP1 [32,42], providing details into the intricate mechanisms that underlie MYC’s control of translation in MM cells. LARP1 has been shown to repress 5′ terminal oligopyrimidine (5’TOP) motif mRNA translation [43,44,45,46], suggesting MYC’s control over translation programs of specific mRNA classes. The translation of 5′TOP transcripts produces ribosomal proteins and several protein-synthesis factors [43,47]. Many different identified MYC target proteins involved in translation (Table 3) are produced by the translation of 5′TOP transcripts [12,48], indicating a mechanism by which MYC selectively affects gene expression. Interestingly, a feedforward loop between MYC and LARP1 was recently identified to promote tumorigenesis in colorectal cancer [49]. In addition to MYC regulation, LARP1 has also been shown to interact with Y-Box Binding Protein 1 (YB-1), another regulator of MYC expression, which itself is encoded by a 5′TOP mRNA [44,49]. YB-1 expression, which was also determined to be downregulated upon MYC depletion (Figure 2 and Appendix A), was already described as MYC-regulated, which in turn controls MYC mRNA translation and was demonstrated to be essential for multiple myeloma cell survival [50]. This suggests an intricate interplay between MYC and MYC target proteins, such as RNA-binding proteins (RBPs), in controlling specific gene expressions.

Using the GO database, we additionally found that the majority of MYC-regulated proteins share RNA-binding functions (GO:0003723), with mRNA processing factors and RBPs being downregulated upon MYC knockout in MM-derived cell lines (Figure 5A,B). Apart from LARP1, the Heterogeneous Nuclear Ribonucleoprotein C (hnRNPC) was further evaluated and was shown to be reduced in MYC-depleted MM cells (Figure 5C,D). Remarkably, hnRNPC was initially known to regulate splicing processes but was later found to modulate MYC mRNA translation during the G2/M cell cycle phase [51,52]. These and our data together reveal a MYC-hnRNPC autoregulatory loop that might be important for MM progression. Indeed, the depletion of hnRNPC by two different sgRNAs negatively affected RPMI8226 and LP1 cell proliferation, indicating its importance for MM cell growth (Figure 6A). In addition, we examined the knockout of LARP1 and its impact on MM cell proliferation. As shown in Figure 6B, LARP1 knockout had minimal effects on proliferation, which was similarly observed for two other mRNA-associated members of the LARP family (LARP4 and LARP4B) [53,54,55]. However, although this implies the presence of functional redundancies of LARP isoforms, by simultaneous depletion of these three LARP proteins (LARP1, -4, and -4B), a considerable proliferation disadvantage was observed (Figure 6C), highlighting the critical role of LARP proteins as indispensable factors in MM cell proliferation.

Altogether, a high expression of the translation initiation proteins (4EBP1 and LARP1) and the RNA processing factor hnRNPC was frequently found in MM, associated with high MYC expression, unfavorable overall survival, and progressive disease stages (Figure 7). This is further supported by previous studies showing that other members of the translation initiation complex are potential therapeutic targets for MM [56,57]. More recently, RNA-binding proteins, including hnRNPC, were found to be prognostic biomarkers in MM and contribute to tumorigenesis through the spliceosome pathway [58]. Thus, our findings shed light on the complex interplay of target pathways that drive MYC-mediated regulation of cellular processes and offer new avenues for the development of targeted therapies for MM.

## 5. Conclusions

In conclusion, our data indicate that the expression of RNA-binding proteins involved in translation is MYC-driven, prognostically unfavorable, and highly dysregulated in MM. As there are currently no effective strategies to target MYC directly, our findings indicate that intervening in certain translational sub-processes might be an effective therapeutic strategy to target MYC dependence in MM. The correlation between MYC expression, MM disease progression, and hnRNPC, as well as LARP1 expression, demonstrates these identified MYC target proteins as putative predictors of MM patient survival and disease progression.

## Figures and Tables

**Figure 1 cancers-15-05508-f001:**
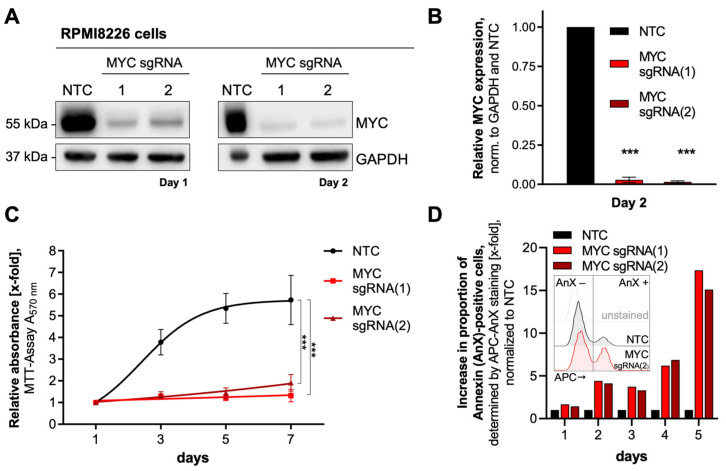
Characterization of MYC knockout RPMI8226 cells. (**A**) Western blots were probed with anti-MYC antibodies to detect MYC expression one day (left panel) and two days (right panel) after transduction with MYC-targeting sgRNAs. NTC RPMI8226 cells served as controls. An anti-GAPDH antibody was used for loading control. (**B**) Quantification of MYC expression on day two is shown in A, right panel. Normalized values are the mean ± SD of three replicates. Statistical significance was determined by one-way ANOVA with Bonferroni’s Multiple Comparison Test; *p* < 0.0001 (***). (**C**) Time-dependent cell viability/metabolic activity, measured by the MTT assay, after MYC knockout. The data are normalized to day 1 and presented as the mean ± SD of five technical replicates. Statistical significance was determined by a one-way ANOVA of the rate constant of an exponential fit excluding the plateau phase at day 7 with Bonferroni’s Multiple Comparison Test, *p* < 0.0001 (***). (**D**) Determination of apoptosis in MYC knockout cells after transduction at indicated timepoints. Multiples of apoptotic cells were measured by AnnexinV (AnX)-APC fluorescence (depicted in the square as an example). The values of each time point were normalized to the respective NTC. Abbreviations: GAPDH, Glyceraldehyde 3-phosphate dehydrogenase; kDa, kilo Dalton; NTC, non-target control.

**Figure 2 cancers-15-05508-f002:**
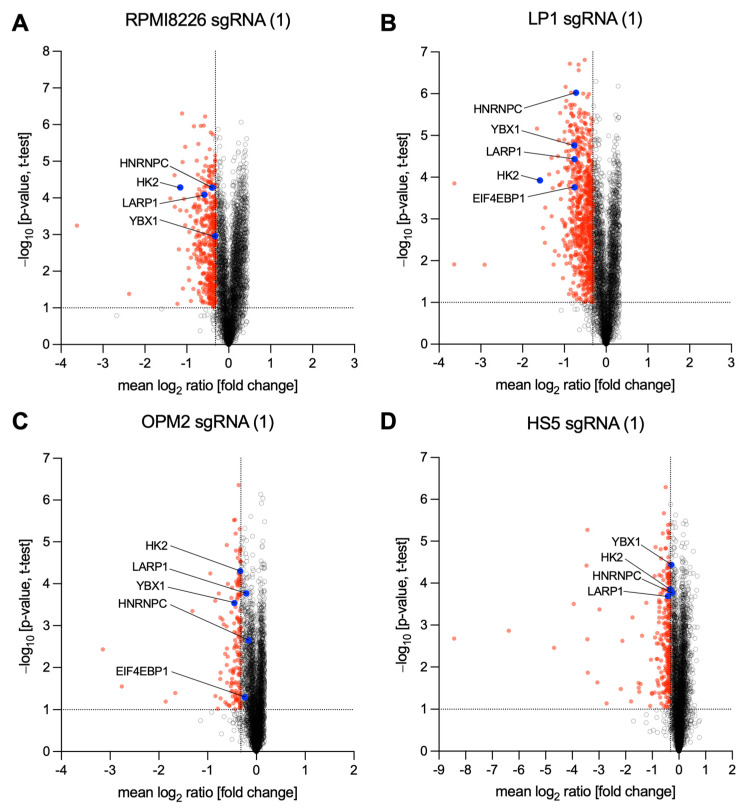
Volcano plots showing fold changes of protein expression upon MYC depletion. (**A**–**D**) Log_2_ [fold change] values and the respective *p*-values (–log_10_ *p*-value, *t*-test, four technical replicates) of quantified proteins two days after transduction by MYC targeting sgRNA(1) in cell lines RPMI8226, LP1, OPM2, and HS5 cells as indicated. Red-marked proteins were significantly downregulated. The thresholds for significant regulations (log_2_ [fold change]) were set according to the publication of known MYC target proteins (described in [27]). Outlier analysis was performed using the Benjamini–Hochberg method with a false discovery rate (FDR) < 0.05. Further analyzed MYC target proteins are labeled in blue.

**Figure 3 cancers-15-05508-f003:**
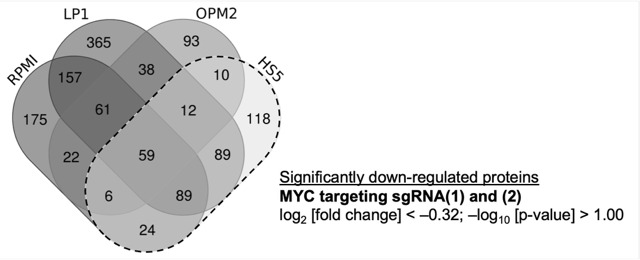
Combined Venn diagram representing total numbers of downregulated proteins by both MYC targeting sgRNA(1) and (2) of four used cell lines. The threshold for protein regulations was set to log_2_ [FC] < −0.32 (0.80-fold, compared to NTC). Statistical significance for protein regulations was determined by *t*-tests (sgRNA targeting MYC vs. NTC; –log [*p*-value] > 1.0). Protein regulation in HS5 cells only (dotted line) was excluded from the protein group being regulated in MM-derived cell lines. Intersections were calculated by the web-based Venn diagram tool (https://bioinformatics.psb.ugent.be/webtools/Venn/, accessed on 15 June 2022).

**Figure 4 cancers-15-05508-f004:**
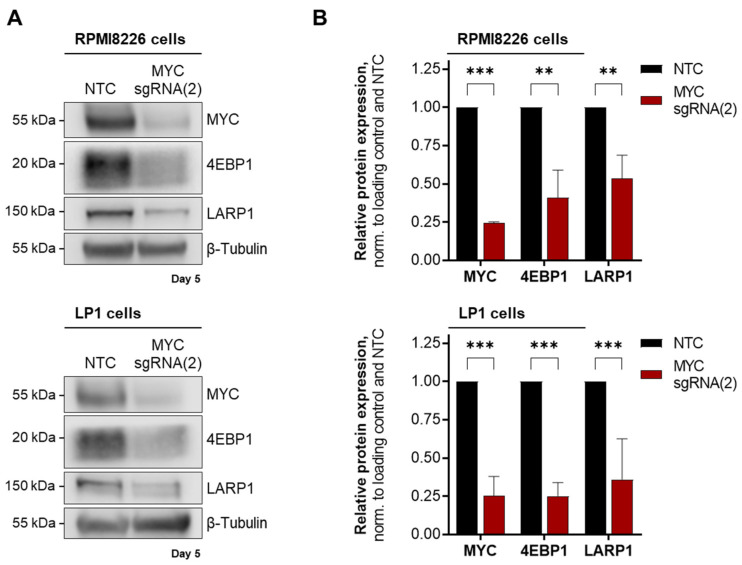
Validation of potentially MYC target proteins in MM-derived cell lines. (**A**) Representative Western blots of downregulated target proteins (4EBP1 and LARP1) involved in translational control in RPMI8226 (upper panel) and LP1 cells (lower panel) on day 5 after transduction with MYC targeting sgRNA(2). β-Tubulin served as loading control. (**B**) Quantification of Western blot analyses. Normalized values are the mean ± SD of two (RPMI8226) and three replicates (LP1). Statistical significance was determined by a two-way ANOVA with Bonferroni’s Multiple Comparison Test; *p* < 0.001 (***), *p* < 0.01 (**). Abbreviation: 4EBP1, 4E Binding Protein; NTC, non-target control; kDa, kilo Dalton; LARP1, La-Ribonucleoprotein 1.

**Figure 5 cancers-15-05508-f005:**
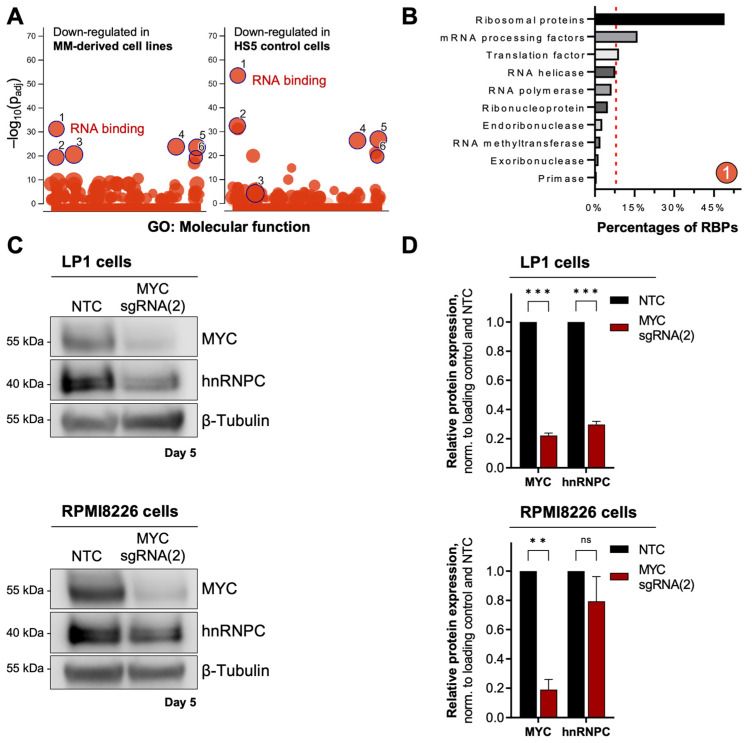
Downregulation of hnRNPC upon MYC depletion. (**A**) Manhattan plot showing molecular function (MF) enrichment analysis of downregulated proteins in MM-derived cell lines (**left** panel) and HS5 control cells (**right** panel). The *x*-axis represents MF terms from GeneOntology (GO), where terms of the same GO subtree are located closer to each other. The *y*-axis shows the adjusted enrichment *p*-values on a negative log_10_ scale. The circle sizes are by the corresponding MF term size. Enrichment analysis was performed using g:Profiler (version e106_eg53_p16_65fcd97) [21] with the Benjamini–Hochberg FDR multiple testing correction method applying a significance threshold of 0.05. Significantly enriched MFs were numbered ((1) RNA-binding, (2) nucleic acid-binding, (3) structural constituent of the ribosome, (4) organic cyclic compound binding, (5) heterocyclic compound binding, and (6) catalytic activity, acting on a nucleic acid). (**B**) Identified downregulated RNA-binding proteins (RBPs, numbered (1) in (A) on day two upon MYC depletion (sgRNA(2)), exemplary shown for LP1 cells. Ribosomal proteins, mRNA processing, and translation factors each comprise over 8% of RBPs (red dotted line). (**C**) Western blot analysis of Heterogeneous Ribonucleoprotein Particle C (hnRNPC) expression in MYC-depleted LP1 (upper) and RPMI8226 cells (lower panel) on day five upon MYC depletion (sgRNA(2)). β-Tubulin served as loading control. (**D**) Quantification of decreased hnRNPC expression shown in C. Normalized values are the mean ± SD of two replicates. Statistical significance was determined by a two-way ANOVA with Bonferroni’s Multiple Comparison Test; *p* < 0.001 (***), *p* < 0.01 (**); ns, non-significant.

**Figure 6 cancers-15-05508-f006:**
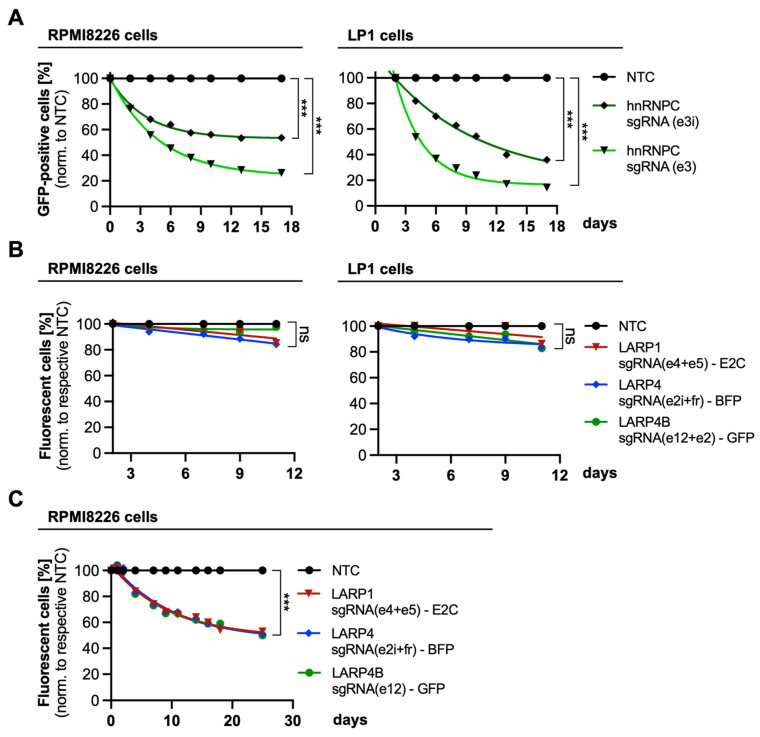
The competitive proliferation of MYC target-depleted MM cell lines. (**A**) Competitive proliferation of hnRNPC-depleted RPMI8226 (**left** panel) and LP1 cells (**right** panel). Measured are percentages of GFP-fluorescent knockout cells on the first day and day 17 of the proliferation assays. (**B**) Competitive proliferation analysis of RPMI8226 (**left** panel) and LP1 cells (**right** panel) upon individual LARP protein knockouts using a combination of two sgRNA targeting one LARP gene. (**C**) Competitive growth analysis of RPMI8226 cells upon triple LARP protein knockout (LARP1, LARP4, and LARP4B). All three LARP proteins were knocked out simultaneously (‘triple KO’), using sgRNA as indicated, but percentages of individual fluorescence signals (E2Crimson (E2C), BFP, or GFP, as indicated) were measured separately. Values were normalized to the respective NTC (triple NTC). Statistical significance was determined by the extra sum-of-squares F test to evaluate differences in one-phase decay curve fitting; *p* < 0.0001 (***); ns, non-significant.

**Figure 7 cancers-15-05508-f007:**
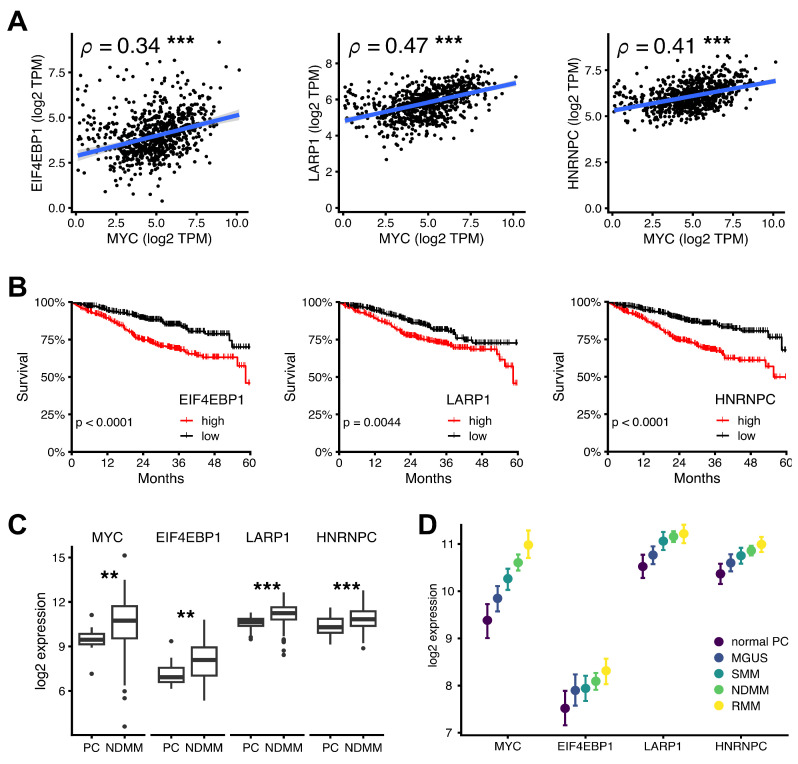
MYC and MYC target mRNA expression and stratification of overall survival in multiple myeloma patients. (**A**) Correlation of gene expression of MYC with EIF4EBP1 (4EBP1), LARP1, and hnRNPC expression (TPM, transcripts per million) in CD138+ bone marrow samples from MMRF CoMMpass study patients at first visit (*n* = 762). Measurements from individual patients (black dots) and robust linear regression lines (blue) are shown. The association between MYC and its target is quantified by the Spearman rank coefficient *ρ* (***, *p*-value < 0.001, calculated using algorithm AS 89). (**B**) Overall survival of MMRF CoMMpass study patients stratified by above- (“high”, *n* = 381) and below-median (“low”, *n* = 381) mRNA expression of EIF4EBP1, LARP1, and hnRNPC. Low- and high-expression groups were compared to the *p* -value by the log-rank test. (**C**) Gene expression of MYC and MYC targets in plasma cells (PC) from healthy donors (*n* = 15) and from newly diagnosed multiple myeloma (NDMM) patients (*n* = 75). *p*-value by the Wilcox signed rank test; *p* < 0.001 (***), *p* < 0.01 (**). (**D**) Parameter estimates and 95% probability intervals for log_2_ expression per gene and disease stage (normal PC, monoclonal gammopathy of unknown significance (MGUS, *n* = 21), smoldering multiple myeloma (SMM, *n* = 23), newly diagnosed multiple myeloma (NDMM, *n* = 24), and relapsed multiple myeloma (RMM, *n* = 28) by ordinal regression analysis.

**Table 1 cancers-15-05508-t001:** Karyotypic abnormalities of MYC in MM cell lines, according to [14,15]. The total chromosome number (Chr) and the number of karyotypically abnormal (A) and normal (N) MYC alleles are listed. VH, variable immunoglobulin (Ig) heavy chain locus; CH, constant region Ig heavy chain.

Cell Lines	Chr	A	N	Abnormality
RPMI8226	60	1	3	MYC insertion on t(16;22)(q23;q11):der(16)
LP1	80	5	1	t(8;14) variant: der(8) with VH
OPM2	67	2	2	CH insertion on t(1;8)(q12?;q24):der(8)

**Table 2 cancers-15-05508-t002:** Oligonucleotide for sgRNAs (5′-3′ orientation). Lowercase letters indicate the gene-specific sequences; uppercase letters indicate the overhangs required for cloning and transcription initiation.

sgRNA ID	5′-3′ Sequences	Fluorescent Marker
MYC (sgRNA(1))	sense: CACCGgacgctgtgcccgcgggcg antisense: AAACcgcccgcgggcacagcgtcC	GFP
MYC (sgRNA(2))	sense: CACCGacgttgcggtcacaccctt antisense: AAACaagggtgtgaccgcaacgtC	GFP
hnRNPC (sgRNA(e3))	sense: CACCGgagtagaggggacggagaa antisense: AAACttctccgtcccctctactcC	GFP
hnRNPC (sgRNA(e3i))	sense: CACCGgttccggacctgagtagag antisense: AAACctctactcaggtccggaacC	GFP
LARP1 (sgRNA(e4))	sense: CACCGaaatcagatgaatcagggg antisense: AAACcccctgattcatctgatttC	E2Crimson (E2C)
LARP1 (sgRNA(e5))	sense: CACCGtatgtgtctaggctccggt antisense: AAACaccggagcctagacacataC	E2Crimson (E2C)
LARP4 (sgRNA(e2i))	sense: CACCGatatttaagacttaccctc antisense: AAACgagggtaagtcttaaatatC	BFP
LARP4 (sgRNA(fr))	sense: CACCGcatcaggtgctcatcctga antisense: AAACtcaggatgagcacctgatgC	BFP
LARP4B (sgRNA(e12))	sense: CACCGcgattcctagtgcagagag antisense: AAACctctctgcactaggaatcgC	GFP
LARP4B (sgRNA(e2))	sense: CACCGgatggaacttgtctgagaa antisense: AAACttctcagacaagttccatcC	GFP
NTC	sense: CACCGttccgggctaacaagtcct antisense: AAACaggacttgttagcccggaaC	GFP, BFP, or E2C

**Table 3 cancers-15-05508-t003:** Identified proteins involved in translation, according to the REACTOME database.

	Putative MYC Target Proteins
MM-derived cell lines (RPMI8226, LP1, and OPM2)	PABPC1, EIF5B, EIF4G1, EIF4EBP1, EIF4B, EIF4A1, EIF3K, EIF3I, EIF3G, EIF3F, EIF3E, EIF1AX + (RPS9, RPS7, RPS6, RPS3, RPS29, RPS27, RPS21, RPS20, RPS18, RPS17, RPS15, RPS14, RPS12, RPS11, RPS10, RPL9, RPL5, RPL38, RPL36AL, RPL23)

## Data Availability

The mass spectrometry proteomics data have been deposited in the MassIVE database with dataset identifier MSV000092169 (https://massive.ucsd.edu). MS data as MaxQuant output, analyzed by Perseus software (v1.6.0.7), are provided in a separate Excel file (Appendix A).

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
