# Peer review of "The MYC-Regulated RNA-Binding Proteins hnRNPC and LARP1 Are Drivers of Multiple Myeloma Cell Growth and Disease Progression and Negatively Predict Patient Survival"

_cancers, 2023, doi:10.3390/cancers15235508_

Round 1
Reviewer 1 Report
Comments and Suggestions for Authors
The study presents a compelling exploration of the roles of MYC-regulated RNA binding proteins, hnRNPC and LARP1, in driving multiple myeloma cell growth and disease progression. The elucidation of their mechanisms provides valuable insights into the underlying molecular processes of this complex hematologic malignancy. The study is well-structured and effectively communicates the research objectives, methods, results, and implications. This article is suitable for the Journal and could be accepted before answering some minor questions.
1.In Figure 2, the author utilizes the color red to indicate significantly changed proteins, coinciding with using the same color to mark the protein names. It is advisable to select alternative colors for protein marking to prevent potential confusion among readers. Additionally, it is suggested to clearly differentiate the protein names from the highlighting, ensuring that the author's markup is easily discernible. The current uniform use of red makes it challenging for readers to distinguish whether the protein names are appropriately marked.
2. Still in Figure 2, the absence of a threshold dotted line needs to be addressed. Furthermore, the distinction between red dashed lines and red hollow circles should be explicitly described for the readers' comprehension
3. P9, line 353. It is suggested that the author modify the threshold value of the change factor from < (-0.32) to 1.5 times, which aligns better with readers' accustomed reading practices.
4. In Figure 5A, it is unclear whether the size of the circle represents significance or relevance. The author is encouraged to provide an explanation for the readers, elucidating the meaning behind the circle size in the context of the depicted data
Author Response
Reviewer #1: The study presents a compelling exploration of the roles of MYC-regulated RNA binding proteins, hnRNPC and LARP1, in driving multiple myeloma cell growth and disease progression. The elucidation of their mechanisms provides valuable insights into the underlying molecular processes of this complex hematologic malignancy. The study is well-structured and effectively communicates the research objectives, methods, results, and implications. This article is suitable for the Journal and could be accepted before answering some minor questions.
We thank Reviewer 1 for this very kind and positive evaluation of our manuscript.
- In Figure 2, the author utilizes the color red to indicate significantly changed proteins, coinciding with using the same color to mark the protein names. It is advisable to select alternative colors for protein marking to prevent potential confusion among readers. Additionally, it is suggested to clearly differentiate the protein names from the highlighting, ensuring that the author's markup is easily discernible. The current uniform use of red makes it challenging for readers to distinguish whether the protein names are appropriately marked.
- Still in Figure 2, the absence of a threshold dotted line needs to be addressed. Furthermore, the distinction between red dashed lines and red hollow circles should be explicitly described for the readers' comprehension
Thank you for this constructive objection. We have now made Figure 2 more readily understandable for the reader. Specifically, we focus on the volcano plots based on sgRNA 1 in the new Figure 2 and have moved the volcano plots based on sgRNA 2 to a new Supplementary Figure S2 as they have the same information content.
In both figures, we show the significantly down-regulated proteins in red, and the proteins that were investigated in this manuscript in blue. We do not show the significantly up-regulated proteins here anymore, since they were not part of this work. Additionally, according to your suggestion, we have introduced dashed lines defining thresholds.
We also avoid the rather confusing use of different threshold levels. In the original manuscript, we presented bioinformatically calculated thresholds in Figure 2. However, in the course of the analysis, we defined our own thresholds based on a literature study of known MYC target proteins ("biological thresholds"), which were then subsequently used in the pathway analysis throughout the manuscript. We now refrain from showing the bioinformatically calculated thresholds and only show the biological thresholds we used. Therefore, the following adjustments have been made in the text:
In Methods (paragraph 2.7), the sentence “Default thresholds were analytically set to identify regulated proteins that showed a minimum of 2-fold up- or down-regulation in each cell line in at least two biological replicates” was deleted and replaced by the following sentence: “Threshold values to identify regulated proteins were adjusted according to measured log2 (fold change) values of previously described MYC target proteins”
In the legend of Figure 2 the description of the threshold levels was changed to: “The thresholds for significant regulations (log2 [fold change]) were set according to publication-known MYC target proteins (described in [25]).“
We think that by doing so, the figure is significantly improved.
- P9, line 353. It is suggested that the author modify the threshold value of the change factor from < (-0.32) to 1.5 times, which aligns better with readers' accustomed reading practices.
Thank you for bringing this to our attention. We have changed this text to: "< (-0.32)” was changed to “< 0.8-fold (=log2 (FC) < (-0.32))". (line 369)
- In Figure 5A, it is unclear whether the size of the circle represents significance or relevance. The author is encouraged to provide an explanation for the readers, elucidating the meaning behind the circle size in the context of the depicted data
Thank you for this valuable comment. To make it clearer what is shown here, we have included in the legend to Figure 5A the information from the web-based g-Profiler software that was used to calculate the results shown here: "The x-axis represents MF terms from GeneOntology (GO), where terms of the same GO subtree are located closer to each other. The y-axis shows the adjusted enrichment p-values in negative log10 scale. The circle sizes are in accordance with the corresponding MF term size".
For better readability of figure legend 5A, we have moved the explanation of the highlighted numbers in figure 5A to the end of figure legend 5A.
Reviewer 2 Report
Comments and Suggestions for Authors
The work by Seibert and colleagues described the use of MYC silencing and SILAC analysis of proteins with decreased synthesis, linking them to MYC function. The authors show in an interesting way that proteins involved in initiation of translation are regulated by MYC and their high expression is associated with poorer prognosis in myeloma, suggesting that more translationally active disease is more aggressive. The manuscript is well written and logic, I have only few comments or suggestions:
1. While the authors focused on proteins that are less abundant upon MYC knock out, there is also a long list of proteins, which are more abundant in MYC-depleted cells. The authors omitted them in this manuscript, however, do they suggest a rescue mechanism for MM to survive without MYC?
2. Figure 1D – please indicate statistical significance and mark where the difference between NTC and sgRNA MYC are significant.
3. Figure 2: The initial thresholds for significant regulations (log2[fold change]) were analytically set to ±1.0 (RPMI8226, sgRNA1), to ±0.9 (RPMI8226, sgRNA(2)), to ±1.1 (LP1, sgRNA(1) and (2)), to ±0.5 (OPM2, sgRNA(1) and (2)), to ±0.6 (HS5, sgRNA(1)) and to ±0.7 (HS5, sgRNA(2)). Why is that, why in each cell line different thresholds for significant changes are set?
4. Line 315: The three MM-derived cell lines RPMI8226, LP1, and OPM2, which were chosen due to expression of both abnormal and normal MYC alleles… - please indicate which cell lines have normal and which abnormal MYC or what it means by abnormal: rearrangements, fusions…? Or does all three cell lines express one wild-type and one mutated allele?
5. Lines 384-387, it is not clear to me why the authors focused first on MM specific proteins and then continued with LARP1 analysis, which showed to be deregulated also in HS5 cells upon MYC knock out, this part should be elaborated.
6. While in paragraph 3.3 the authors say that MYC-driven translation is deregulated in MM, they hypothesize it from deregulated levels of key translation factors. It is also important to show, for example using a SunSet assay, that translation is indeed impaired in MYC-knock out cells.
7. Figure 6 – there is no information what the data represent, only single experiment if there are no SD values? In case more biological repeats were performed, a statistical significance should be indicated.
Author Response
Reviewer #2: The work by Seibert and colleagues described the use of MYC silencing and SILAC analysis of proteins with decreased synthesis, linking them to MYC function. The authors show in an interesting way that proteins involved in initiation of translation are regulated by MYC and their high expression is associated with poorer prognosis in myeloma, suggesting that more translationally active disease is more aggressive. The manuscript is well written and logic, I have only few comments or suggestions:
We are grateful to the reviewer for this positive evaluation of our manuscript.
- While the authors focused on proteins that are less abundant upon MYC knock out, there is also a long list of proteins, which are more abundant in MYC-depleted cells. The authors omitted them in this manuscript, however, do they suggest a rescue mechanism for MM to survive without MYC?
The intention of this work was to identify potential therapeutically targetable proteins that could be used in place of the non-targetable MYC in patients. Therefore, we did not focus on proteins that are upregulated after MYC KO in his study. However, we agree with the reviewer that these proteins are highly interesting from a basic research point of view and may contribute to the understanding of the molecular mechanism of MYC. However, as suggested by the reviewer, these proteins may be proteins upregulated by secondary effects to cope of MYC depletion or also proteins regulated in the early phase of apoptosis.
To clarify in our manuscript that we have not focused on upregulated proteins in this study, we have removed the highlighting of the up-regulated proteins in the the new Figure 2 and Supplementary Figure 2 (adjusted according to the suggestions of Reviewer 1) and concentrate on the down-regulated proteins only.
- Figure 1D – please indicate statistical significance and mark where the difference between NTC and sgRNA MYC are significant.
The analysis of apoptosis induction to identify an optimal time point for the SILAC screening after MYC knockout shown here, was performed only once, therefore a statistical evaluation of this graph is not possible. Considering the short half-life of MYC and thus the possibility of very early apoptosis induction after MYC depletion, we initially assessed when we exceed a threshold of 5% of apoptotic cells in culture (Figure 1D, 72 hours post transduction). To be on the safe side of detecting direct MYC target proteins and reduce the amount of proteins regulated by secondary effects such as apoptosis, we therefore harvested our samples for mass-spectrometric analysis 48 hours post transduction. Thus, the apoptosis assay was only used to make sure that the majority of the cells are not already apoptotically primed. To clarify our intention, we added the sentence “a time point at which the 95% of cells are viable despite MYC depletion.” to line 325.
- Figure 2: The initial thresholds for significant regulations (log2[fold change]) were analytically set to ±1.0 (RPMI8226, sgRNA1), to ±0.9 (RPMI8226, sgRNA(2)), to ±1.1 (LP1, sgRNA(1) and (2)), to ±0.5 (OPM2, sgRNA(1) and (2)), to ±0.6 (HS5, sgRNA(1)) and to ±0.7 (HS5, sgRNA(2)). Why is that, why in each cell line different thresholds for significant changes are set?
Thank you for bringing this admittedly somewhat confusing point to our attention. The threshold levels shown here are bioinformatically calculated and result from an outlier analysis combined with a t-test of each individual experiment, therefore each experiment also has its own significance threshold. To avoid this confusion, we have removed the bioinformatically determined levels from this manuscript and only show and mention those used in the pathway analysis, which we determined by looking at known MYC target proteins (“biological threshold”) and according to these proteins setting the significance level for all experiments. We have modified the respective text passages accordingly. In the passage you have highlighted, we have replaced the text with: "The thresholds for significant regulations (log2 [fold change]) were set according to publication-known MYC target proteins (described in [25]).“
- Line 315: The three MM-derived cell lines RPMI8226, LP1, and OPM2, which were chosen due to expression of both abnormal and normal MYC alleles… - please indicate which cell lines have normal and which abnormal MYC or what it means by abnormal: rearrangements, fusions…? Or does all three cell lines express one wild-type and one mutated allele?
This aspect indeed needs to be indicated. I thank you for bringing it to our attention. We have included a table in this regard (table 1) where we show the characteristics of the cell lines in terms of the number of normal and translocated alleles. These data are taken from the cited references [14,15]. Due to rearrangement of cited references, all references are shifted from Reference #14
- Lines 384-387, it is not clear to me why the authors focused first on MM specific proteins and then continued with LARP1 analysis, which showed to be deregulated also in HS5 cells upon MYC knock out, this part should be elaborated.
In the pathway analysis we performed using REACTOME, a cluster of proteins emerged that was classified in the Translational Initiation Pathway. In this pathway, eukaryotic initiation factor family members (EIFs) were found with 4EBP1 as a known MYC target protein. In further experiments, we then focused our attention on proteins of this pathway independent of their regulation in HS5 cells. However, when reviewing other regulated proteins, we noticed that LARP1 is also a translation initiation factor but is not part of this pathway in REACTOME. In REACTOME, LARP1 is only found in the classification "Covid-19 pathways". We therefore added LARP1 protein to our analyses. although it is not correctly classified in REACTOME. To clarify this somewhat confusing procedure, we added the following sentence at the first mention of LARP1: “…but is not part of the translation initiation pathway defined in REACTOME.” line 414.
- While in paragraph 3.3 the authors say that MYC-driven translation is deregulated in MM, they hypothesize it from deregulated levels of key translation factors. It is also important to show, for example using a SunSet assay, that translation is indeed impaired in MYC-knock out cells.
Thank you for raising this point.
We believe that there are many examples in the literature showing that MYC is involved in translational control, so we considered this point as current knowledge, which has been sufficiently mentioned in the following publications; cited in our manuscript:
[12] Ruggero, D. The role of Myc-induced protein synthesis in cancer. Cancer Res. 2009, 69, 8839. doi:10.1158/0008-5472.CAN-09-1970.
[27] Van Riggelen, J.; Yetil, A.; Felsher, D.W. MYC as a regulator of ribosome biogenesis and protein synthesis. Nat. Rev. Cancer 2010, 10, 301–309. doi:10.1038/NRC2819.
[28] Schmidt, E. V. The role of c-myc in regulation of translation initiation. Oncogene 2004 2318 2004, 23, 3217–3221. doi:10.1038/sj.onc.1207548.
Similar assays as the requested Sunset assay have been shown for example in Morcelle et al., 2019 (doi: 10.1158/0008-5472.CAN-18-2718.) we added this publication to the discussion: “Conversely, depletion of MYC in colorectal tumors results in significant impairment of global translation [33].”
- Figure 6 – there is no information what the data represent, only single experiment if there are no SD values? In case more biological repeats were performed, a statistical significance should be indicated.
Because of the very clear results obtained here, this experiment was performed only once. To ensure significance, an extra sum-of-squares F-test was performed and corresponding symbols were introduced into the figure. The following sentence was introduced into the legend of Fig. 6: "Statistical significance was determined by Extra sum-of-squares F test to evaluate differences in one-phase decay curve fitting; p<0.0001 (***); ns, non-significant."
Round 2
Reviewer 2 Report
Comments and Suggestions for Authors
The manuscript has been improved and I have no more suggestions.